# E-Selectin and Asymmetric Dimethylarginine Levels in Adult Cyanotic Congenital Heart Disease: Their Relation to Biochemical Parameters, Vascular Function, and Clinical Status

**DOI:** 10.3390/cells13171494

**Published:** 2024-09-05

**Authors:** Sonia Alicja Nartowicz, Ludwina Szczepaniak-Chicheł, Dawid Lipski, Izabela Miechowicz, Agnieszka Bartczak-Rutkowska, Marcin Gabriel, Maciej Lesiak, Olga Trojnarska

**Affiliations:** 11st Department of Cardiology, Poznan University of Medical Sciences, 61-701 Poznań, Poland; abartczak-rutkowska@ump.edu.pl (A.B.-R.); maciej.lesiak@skpp.edu.pl (M.L.); olga.trojanrska@skpp.edu.pl (O.T.); 2Department of Hypertensiology, Angiology and Internal Medicine, Poznan University of Medical Sciences, 61-701 Poznań, Poland; lszczepaniakchichel@ump.edu.pl (L.S.-C.);; 3Department of Computer Science and Statistics, Poznań University of Medical Sciences, 61-701 Poznań, Poland; iza@ump.edu.pl; 4Department of General and Vascular Surgery, Poznan University of Medical Sciences, 61-701 Poznań, Poland; mgabriel@ump.edu.pl

**Keywords:** cyanotic congenital heart disease, cyanosis, augmentation pressure, flow-mediated dilatation, endothelium, ADMA

## Abstract

**Background and Aim:** Patients with cyanosis secondary to congenital heart disease (CHD) are characterized by erythrocytosis and increased blood viscosity, which contribute to endothelial dysfunction, increased arterial stiffness, and impaired vascular function, which may affect the final clinical presentation. Asymmetric dimethylarginine (ADMA) and e-selectin (e-sel) are valuable biomarkers for endothelial and vascular dysfunction. Their concentration levels in blood serum have the potential to be an accessible tool that reflects the severity of the disease. We aimed to assess e-sel and ADMA levels and their relationship with the clinical status and endothelial and vascular function. **Methods:** A cross-sectional study, including 36 adult CHD cyanotic patients [(17 males) (42.3 ± 16.3 years)] with an arterial blood oxygen saturation less than 92% and 20 healthy controls [(10 males) (38.2 ± 8.5 years)], was performed. All the patients underwent a clinical examination, blood testing, and cardiopulmonary tests. Their endothelial function was assessed using the intima media thickness and flow-mediated dilatation. Vascular function, using applanation tonometry methods, was determined using the aortic systolic pressure, aortic pulse pressure, augmentation pressure, augmentation index, pulse pressure amplification, and pulse wave velocity. **Results:** The concentrations of e-sel and ADMA were significantly higher in the patients with CHD. The E-sel levels correlated positively with red blood cells, hemoglobin concentration, hematocrit, and augmentation pressure; they correlated negatively with blood oxygen saturation, the forced expiratory one-second volume, forced vital capacity, and oxygen uptake. The ADMA levels were found to correlate only with age. **Conclusions:** The E-sel level, unlike ADMA concentration, reflects the severity of erythrocytosis and hypoxia and, thus, the physical status of patients with cyanotic CHD.

## 1. Introduction

Cyanosis secondary to congenital heart disease (CHD) is a clinical condition characterized by a decrease in blood saturation below 92% and an increase in the concentration of deoxygenated hemoglobin above 5 g/dL in the capillary blood, often accompanied by a blue discoloration of the skin and mucous membranes [1,2]. It may result from primary cyanotic defects, in which, due to the narrowing of the outflow tract of the right ventricle or pulmonary trunk, there is reduced blood flow through the pulmonary bed and, consequently, a decrease in blood saturation. One of the most common defects is the Tetralogy of Fallot and, less frequently, pulmonary atresia or Ebstein’s anomaly. As a result of leakage defects, which include interatrial defects (ASDs), interventricular defects (VSDs), and patent ductus arteriosus, there is increased blood inflow to the pulmonary vessels, which consequently leads to an increase in their resistance, impaired oxygen exchange, and increased pulmonary pressure, which exceeds the system values and leads to reverses of the shunt. Deoxygenated blood enters systemic circulation, significantly reducing oxygen saturation. There are also complex anatomical anomalies, such as hearts with single ventricle physiology, in which cyanosis results from both mechanisms. The unphysiological phenomenon of a lowering blood saturation causes compensatory processes, the most important of which is an increase in the concentration of erythropoietin produced by the kidneys, which increases the production of red blood cells [3]. The resulting growth in the hematocrit increases blood viscosity, decreases nitric oxide secretion, and affects endothelial and vascular function [3]. A consequence of these phenomena is the impairment of the functions of most organs, including the myocardium. Multi-organ complications are also intensified by thromboembolic events resulting from endothelial pathology and the dysfunction of coagulation pathways [4]. Studies of patients with coronary artery disease, generalized atherosclerosis, pulmonary hypertension, or heart failure have shown that increased levels of the vasodilator asymmetric dimethylarginine (ADMA) have been observed in patients with impaired arterial endothelial function. This amino acid is a strong inhibitor of the bioavailability of nitric oxide, a factor that plays a vital role in regulating endothelial function. It is formed as a result of protein hydrolysis, as a result of the activity of specific methyltransferases, the expression of which, on endothelial cells, increases in response to mechanical stress and shear forces, phenomena that are strongly expressed in patients with erythrocytosis and increased viscosity during the course of cyanosis. The mentioned impairment of rheology participates in the activity of e-selectin (e-sel)—a protein specific to endothelial cells, responsible for retaining the leukocytes that flow in the bloodstream on the inner surface of the vessels. This facilitates their penetration into the inner membrane of the blood vessels which, in turn, intensifies the inflammation involved in the process of atherogenesis [5,6,7,8]. As an adhesion molecule, it can also bind to fibrin [9]. There are few studies in the available literature regarding the concentration and clinical significance of the biomarkers mentioned above in patients with cyanosis resulting from CHD [10,11]. This study aims to assess the concentration of e-selectin and ADMA and demonstrate their potential relationship with the function of the endothelium and blood vessels, as well as with the clinical presentation of this population.

## 2. Materials and Methods

### 2.1. Studied Group and Control Group

Thirty-six patients (19 females) with cyanosis resulting from CHD, aged 20–72 (mean age 42.3 ± 16.3 years), with less than 92% arterial blood oxygenation, were enrolled. All the patients were under the care of the Adult Congenital Heart Disease Outpatient Clinic. The control group comprised 20 healthy individuals (10 females) aged 19–64 (mean age 38.2 ± 8.5). Each patient was examined during one visit in the morning before the morning pharmacotherapy dose. The baseline characteristics of all the included individuals are presented in Table 1, Table 2 and Table 3. The exclusion criteria were the coexistence of Down syndrome, acute and chronic inflammatory disease (in the preceding 3 months), concomitant malignancies, diabetes mellitus, and smoking cigarettes. The local bioethical committee approved the study (decision 325/14), and informed consent was obtained from all the patients following the ethical guidelines in the Declaration of Helsinki of 1964.

### 2.2. Laboratory Method and Blood Sampling

Blood samples were collected from the patients in the morning, at least 10 h after the last meal, then centrifuged and frozen at −80 degrees Celsius until assayed. ADMA and e-selectin levels were measured using enzyme-linked immunosorbent assay (ELISA) kits: ADMA—Immunediagnostik AG, Bensheim, Germany; E-selectin Bioassay Technology Laboratory, Shanghai, China. ADMA and e-selectin test disclosed values as low as 0.05 umol/L. 

### 2.3. Vascular and Endothelial Function

The flow-mediated dilatation (FMD) and intima media thickness (IMT) measurements were assessed as previously described [12]. We analyzed the endothelial function of conduit vessels via ultrasound (7–12 MHz linear array transducer, Logic 7, GE) assessment of the brachial diameter changes during artery flow changes. An image of a 5 cm length of the brachial artery in a longitudinal section above the antecubital fossa was performed. We collected baseline parameters like brachial artery diameter and Doppler velocities. For 3 min, a distal-occluding forearm cuff placed just below the antecubital fossa was inflated to 50 mm Hg above the systolic pressure. Brachial artery scans were assessed 120 s after the cuff deflation, including a repeated flow velocity recording for the first 15 s after the cuff release. The vessel diameter response to reactive hyperemia (FMD) was calculated. The percentage change relative to the diameter measured immediately before the cuff inflation was also obtained. We used Logic 7, GE ultrasound machine (Fairfield, CT, USA) with the 7–12 MHz transducer to perform left and right carotid artery scans. We analyzed diastolic images of the best demonstrated IMT with simultaneous visualization of the common carotid artery’s near and far walls. Images were assessed by a single researcher (blinded to the other results) using a quantitative analysis package (Siemens, Berlin, Germany), giving an axial resolution of 0.001 mm.

All patients were tested in resting supine positions between 8.00 and 10.00 a.m. in a temperature-controlled room (20–22 C). A single observer interpreted scans. IMT was quantitatively analyzed with package (Siemens, Berlin, Germany), giving an axial resolution of 0.001 mm.

The central aortic function was assessed using arterial wall stiffness and pulse wave analysis performed with applanation tonometry (Sphygmo- Cor PVx, Version 8.0, AtCor, Sydney, Australia) and estimated by a single observer. To derive the aortic systolic pressure (AoSP), aortic diastolic pressure (AoDP), as well as augmentation pressure (AP), and augmentation index (AI), the central aortic pulse pressure waveform was determined from radial to aortic generalized validated transfer function [13,14]. The AP was calculated as the difference between the second and the first peak of the central aortic pulse wave. The AI is the ratio of the AP to aortic pulse pressure (APP) expressed as a percentage [13,14]. 

Pulse wave velocity (PWV) was assessed as previously described [12]. The transit time was calculated as the time between the foot of the pulse wave and the R wave in a concurrently recorded electrocardiogram. The average of the consecutive beats per 10 s was derived as pulse transit time. The difference between those two transit times was the delay time (t). We measured the distance (d) traveled by the pulse wav22e over the body surface at a distance between the two locations. PWV was calculated as PWV = d/t [m/s] by the SphygmoCor device (SphygmoCor PVx, Version 8.0, AtCor) [13,14]. One examiner performed the assessment.

### 2.4. Cardiopulmonary Test

All included patients performed a maximum, symptom-limited (fatigue and/or dyspnea) cardiopulmonary test on a treadmill according to modified Bruce protocol (Bruce protocol with stage 0 added—3 min at 1.7 km/h, 5% grading), while in the control group, the standard Bruce protocol was used. During each test, gas analysis (Sensor Medics, model Vmax29) with the measurement of the peak oxygen consumption (peak VO2), carbon dioxide production (VCO2), and minute ventilation (VE) was performed. A respiratory quotient (RQ) greater than one was sought for all patients. A standard 12-lead electrocardiogram was continuously recorded. Blood pressure was measured every two minutes using a cuff sphygmomanometer. Peak VO2 was measured, reflecting an average value during the last 20 s of exercise, which was presented as a percentage of predicted peak oxygen consumption and as mL/kg/min, mL/min. The ventilation/carbon dioxide slope (VE/VCO2 slope) was automatically calculated. The spirometric study was performed on every patient just before the cardiopulmonary exercise test. The obtained values were expressed as a percentage of predicted average values, adjusting for age, sex, and body mass.

### 2.5. Statistical Analysis

Statistical analysis was performed using the Statistica 13.3 Programme. Data were presented as the mean value with standard deviation (±SD) and range of minimal-maximal value with a median. We performed the statistical analysis with the t-test for unpaired samples when the variables followed a normal distribution and the Mann–Whitney U test when the variables did not follow a normal distribution. The relationship between variables was determined using the Spearmen rank order or Pearson correlation. The chi-square test was used to compare categorical variables. We considered the results of *p* < 0.05 as statistically significant. 

## 3. Results

### 3.1. Clinical, Vascular, and Cardiopulmonary Exercise Test Parameters: Cyanotic vs. Control Group

As shown in Table 1, a comparison of basic clinical data revealed that cyanotic patients were characterized as typical for their state elevated level of hematocrit (HCT: 54.7 ± 10% vs. 40 ± 3%; *p* < 0.001), hemoglobin (HGB: 11.16 ± 1.62 mmol/L vs. 8.6 ± 0.8 mmol/L; *p* < 0.001), and erythrocytes (RBC: 5.97 ± 1.09 × 1012/L vs. 4.4 ± 0.5 × 1012/L; *p* < 0.001). We did not demonstrate any difference between the study and control groups in analyzing other biochemical parameters, such as the cholesterol level and its fractions, glucose, or creatinine. We showed a significant difference between the concentrations of e-selectin (e-sel: 57.6 ± 15.4 vs. 15.5 ± 8.6; <0.001) and ADMA (1.4 ± 0.5 vs. 0.4 ± 0.1; *p* < 0.01), which were significantly higher in the group of cyanotic patients than in controls. Parameters of the central aortic pulse wave analysis, AP (10.14 ± 7.3 mmHg vs. 5.3 ± 3.3 mmHg; *p* = 0.002) and AI (24.75 ± 13.49 vs. 15.7 ± 8.8%; *p* = 0.01), were higher in the studied group than in the controls (Table 2). All subjects with cyanosis secondary to a CHD were characterized by significantly lower FVC values (3.4 ± 1.1 vs. 4.5 ± 0.7; *p* < 0.001), FVC% (86 ± 16.7 vs. 98.2 ± 8.8; *p* = 0.04), FEV1 (2.5 ± 0.8 vs. 3.6 ± 0.6; <0.001), FEV1% (76.4 ± 19.4 vs. 97.9 ± 9.1; *p* < 0.001), VE/VCO3 (43.0 ± 12.7 vs. 27.7 ± 4.5; *p* < 0.001), VO2 (9.95 ± 8.2 vs. 34.4 ± 6.5 mL/kg/min; *p* < 0.001), and VO2 (48.2 ± 16.7 vs. 48.2 ± 16.7%; *p* < 0.001) than in the control group (Table 3).

### 3.2. Correlation of E-Selectin and ADMA with Analyzed Clinical Parameters 

The analysis performed on the group of patients with cyanosis showed a statistically significant, negative correlation between the e-selectin concentration (*p* < 0.01; R = −0.4) and blood oxygen saturation and a positive correlation of the e-selectin concentration with RBC (*p* < 0.01; R = 0.6), HCT (*p* < 0.01; R = 0.6), and Hgb (*p* < 0.01; R = 0.6) (Table 4, Figure 1). AP was the only vascular parameter that correlated positively with e-selectin (*p* = 0.05; R = 0.3) (Table 4). Additionally, we proved that the e-selectin concentration correlated with the parameters assessed during cardiopulmonary tests: FVC (R = −0.4; *p* = 0.03), FEV1 (R = −0.5; *p* = 0.01), VE/VCO3 (R = 0.5; *p* = 0.003), VO2 (R = −0.4; *p* = 0.02), and VO2% (R = −0.4, *p* = 0.04) (Table 4).

ADMA concentrations correlated with age (R = 0.4; *p* = 0.02) only. No other statistically significant correlations were found.

## 4. Discussion

### 4.1. E-Selectin

Our study showed an increased concentration of e-selectin in the blood serum of examined patients with cyanosis secondary to CHD in the adult population. This protein, located on the surface of the endothelium, is responsible for retaining leukocytes flowing in the bloodstream on the inner surface of the vessel, which consequently intensifies the inflammatory process [5,6,7,8]. The literature describes a close relationship between the e-selectin concentration and the degree of vascular damage in patients suffering from cardiovascular diseases [15,16,17]. At the same time, clinical observation of cyanotic patients indicates generalized multi-organ damage, in which the functioning of the microcirculation is essential. Our analysis seems to confirm this thesis, especially since a pathophysiological relationship has been proven between the substance stimulating the expression of e-selectin and another inflammatory cytokine—TNF-a (Tumor Necrosis Factor alpha), the secretion of which is stimulated, among others, by the profound hypoxemia characteristic for this population [18].

In the available literature, only one study by Smadja et al. assessed the e-selectin concentration in patients with low oxygen saturation resulting from CHD [10]. However, in a group of twenty-six children, these researchers did not demonstrate a relationship between the concentration of the described protein and the degree of blood oxygenation. This is a different outcome from the result obtained in our study, in which this relationship is significant. We also confirmed the close dependence of the e-selectin concentration on parameters indicating erythrocytosis, which characterizes cyanotic patients, suggesting the importance of this phenomenon in the process of e-selectin secretion. This thesis was confirmed by Musolino et al., who analyzed a group of patients with chronic myeloproliferative syndromes in which the concentration of this biomarker was elevated, especially in the case of thromboembolic complications [9]. The above data allow us to conclude that hypoxia and erythrocytosis typical of cyanotic CHD patients favor the expression of e-selectin, a protein that enhances atherogenesis. 

Our study showed that parameters characterizing vascular and endothelial function (PWV, IMT, and FMD) are worse in patients with CHD than in the control group. Still, the differences between the groups did not reach statistical significance. Tarp et al. and Pedersen presented similar conclusions regarding endothelial function (FMD and IMT) in adult patients with cyanosis resulting from CHD [19,20]. However, other authors showed the opposite outcome, indicating a significant impairment of endothelial dysfunction [21,22,23]. Therefore, they investigated smaller populations, while Oeschlin only used a different method, namely, plethysmography of the venous system. Unlike the broad population of patients with cardiovascular diseases, our analysis did not show any relationship between any of the parameters determining the function and structure of vessels and the concentration of e-selectin [5,15,17,24,25]. Similar observations of vascular function (preserved PWV) were presented by Tomkiewicz-Pająk et al. [26]. The authors conducted research on patients after the Fontan procedure, those whose preoperative period was characterized by cyanosis [26]. The severity of this deviation was positively correlated with the age of the procedure, which may suggest that the longer the exposure to hypoxia, the worse the vascular function [26].

Moreover, we proved significantly higher values of AP and AI, an indirect biophysical method of endothelium status assessment using applanation tonometry. AI and AP are indexes of arterial wave reflection, which tend to increase as reflected waves arrive earlier in systole. It can result from increased arterial stiffness (similar to PWV) in the general population or endothelial dysfunction related to hypoxia and changes in rheological properties of blood in cyanosis, as in the case of patients with CHD. This is confirmed by the relationship between AP and e-selectin in our study—a parameter related to endothelial dysfunction. However, we could not establish such a relationship for vascular stiffness parameters applicable to the general population—PWV and IMT or FMD. Our research is a step toward understanding the phenomena governing vascular function in the population of CHD patients, especially the highest-risk group, i.e., patients with cyanosis. Based on an extensive analysis of the literature on endothelial dysfunction in patients with severe CHD (Tetralogy of Fallot, a single ventricle after the Fontan procedure), Sandhu et al. demonstrated higher vascular stiffness, expressed by PWV and AP, than in the healthy population [27]. The author did not distinguish patients with cyanosis. However, the conclusions emphasize the difficulty of potential generalizations and clear findings in such an anatomically non-homogeneous group and the diverse methodology used for vessel assessment. Moreover, in a group of 1125 adult patients with preoperative cyanosis due to various anatomical cardiac anomalies, analyzed by Muller et al., higher vascular stiffness by assessing AP was reported [28]. None of the studies cited above investigated the relationship between vascular stiffness and e-selectin. Such a relationship was sought in extensive studies of patients with cardiovascular diseases resulting from atherosclerosis, but it was not confirmed, which suggests the lack of simple, unambiguous pathophysiological relationships of this disease process [29,30,31].

The physical capacity of our patients with cyanosis resulting from CHD was significantly lower than in healthy people. All the analyzed spiroergometric test results, except heart rate, showed a significant, negative correlation with the e-selectin concentration. This indirectly proves that factors damaging blood flow and tissue oxygen supply determine their proper function. A reduced oxygen supply is also the result of reduced oxygen exchange in the lungs, so desaturation is a common feature of our patients with cyanosis and those suffering from pulmonary diseases. The study of exercise capacity in patients with chronic obstructive pulmonary disease, systemic sclerosis, and bronchiectasis showed an inverse relationship between e-selectin values and the forced expiratory volume in 1 s, forced vital capacity, and lung diffusing capacity [32,33,34], which additionally suggests its importance in the pathogenesis of vascular changes and their clinical consequences in states of hypoxia. To the best of our knowledge, the importance of e-selectin for exercise capacity in the population of patients with CHD has not yet been described.

### 4.2. ADMA

Our study showed that the level of ADMA in cyanotic patients is significantly raised. The concentration of this amino acid increased with the age of patients, which is explained by a more prolonged exposure of the endothelium to inflammatory factors damaging it [35]. Over time, the resulting suppression of the secretion and availability of nitric oxide intensifies the process of endothelial damage and vascular stiffness, which is part of the general aging process [36]. This phenomenon is consistent with the clinical observation of the early appearance of vascular complications in cyanotic patients with CHD [27]. However, our study did not find a relationship between the ADMA concentration and saturation or parameters indicating erythrocytosis, so we do not confirm this thesis.

Similarly to our study, elevated ADMA concentrations are described in three analyses based on hemodynamic tests conducted in a population of patients with pulmonary hypertension, and therefore cyanosis, in the course of CHD [37,38]. The concentration of this amino acid correlated with the values of pulmonary resistance, the increase in which is synonymous with excessive mechanical stress of the blood flow, especially in the presence of increased blood volume due to bronchopulmonary collaterals and raised blood viscosity in cyanosis. Nevertheless, our study did not demonstrate a relationship between the ADMA concentration and the parameters we examined characterizing the function of the endothelium and vessels of systemic arteries. This outcome is confirmed by Malle et al., who obtained similar results as one of the few people analyzing this problem in patients with arterial hypertension [39]. Differently, in the healthy cohort and patients with coronary artery disease, it was observed that an increase in the discussed amino acid is associated with a deterioration of endothelial function assessed by FMD [40,41,42].

In our study, we also did not demonstrate a relationship between the concentration of ADMA and spiroergometric parameters indicating the patient’s physical capacity, which is consistent with the observation carried out in the general population by Selejflot et al. [43]. The relationship between the described amino acid concentration and the degree of cardiac performance was demonstrated by Liu and Potocnjak et al. [44,45]. The only study available in the literature regarding the importance of ADMA in a population of 94 patients with CHD without cyanosis showed an inversely proportional correlation between the concentration of this biomarker and spiroergometric indicators of exercise capacity [46]. Therefore, this problem in the population of cyanotic patients requires more extensive research.

## 5. Conclusions

Increased concentrations of e-selectin and ADMA characterize patients with cyanosis secondary to congenital defects. A causal relationship between desaturation and the resulting erythrocytosis, as well as a consequential relationship in the form of endothelial and vascular damage and impaired exercise capacity, was proven only for the concentration of e-selectin. The ADMA concentration is only related to the patient’s age.

### Limitations

Potential limitations of our study include the small number of patients and their diversity in cardiac anatomy. The low extent of our sample does not allow us to draw any definitive conclusions. However, these limitations are present in most studies analyzing patients with congenital heart defects. Moreover, cyanotic congenital heart defects are the primary cause of cyanosis in some of the study populations. In contrast, in the remaining cases, it is generated by secondary pulmonary hypertension due to shunt defects, the pathophysiology of which is slightly different.

## Figures and Tables

**Figure 1 cells-13-01494-f001:**
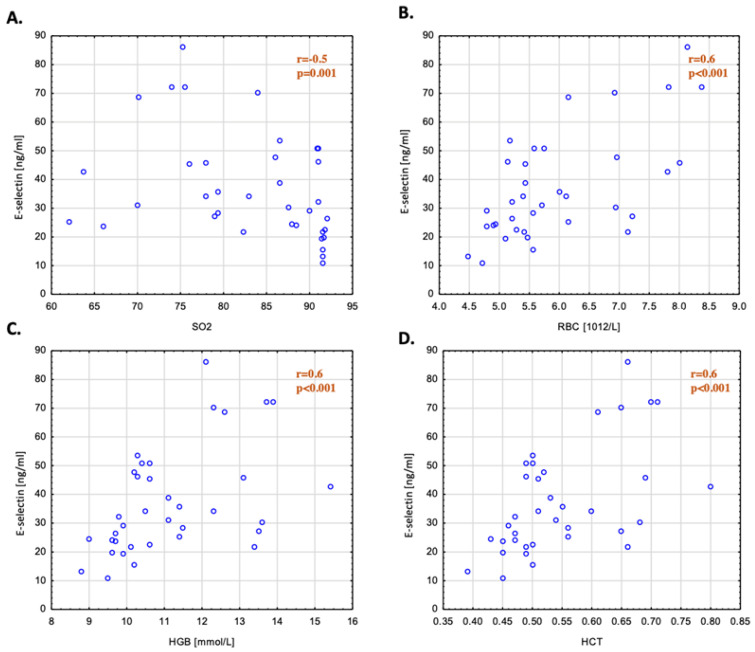
Correlation between e-selectin concentrations and (**A**) SpO2, (**B**) RBC, (**C**) Hgb, and (**D**) HCT values in patients with cyanotic congenital heart disease. Hgb—hemoglobin; HCT—hematocrit; RBC—red blood cells; SpO2—oxygen saturation. Hemoglobin (r = 0.6; *p* < 0.001); hematocrit (r = 0.6; *p* < 0.001); red blood cells (r = 0.6; *p* < 0.001); oxygen saturation (r = −0.5; *p* = 0.001).

**Table 1 cells-13-01494-t001:** Baseline characteristics of the study population and the control group. Data are presented as the mean ± SD, median, and min-max value. cc-TGA—congenitally corrected transposition of the great arteries; HDL—high-density lipoprotein; LDL—low-density lipoprotein; MAPCA’s—major aortopulmonary collateral arteries; RBC—red blood cell count.

	Cyanotic Patients (*n* = 36)	Control Group (*n* = 20)	Cyanotic Patients vs. Control Group (*p*-Value)
Male:Female	17:19	10:10	-
Age (years)	42.33 ± 16.27 41 (24–72)	38.2 ± 8.5 32.5 (25–58)	0.3
Oxygen blood saturation (%)	82.93 ± 8.92 86.3 (62–91.5)	98.2 ± 0.8 98 (96–99)	<0.001
Hematocrit (%)	54.7 ± 10 50 (20–72)	40 ± 3 40 (30–50)	<0.001
Hemoglobin (mmol/L)	11.16 ± 1.62 10.6 (8.8–15.4)	8.6 ± 0.8 8.4 (7.2–10.2)	<0.001
RBC (10^12^/L)	5.97 ± 1.09 4.3 (2.6–7.4)	4.4 ± 0.5 4.39 (3.6–5.4)	<0.001
Total cholesterol (mmol/L)	4.49 ± 1.21 4.4 (2.6–7.6)	5.5 ± 1.1 5.29 (3.95–7.1)	0.7
LDL cholesterol (mmol/L)	2.75 ± 1.06 2.6 (1.4–6)	3.1 ± 0.6 3.1 (2.1–4.3)	0.3
HDL cholesterol (mmol/L)	1.28 ± 0.3 1.2 (0.75–1.98)	1.4 ± 0.3 1.3 (1–1.9)	0.5
Glucose (mmol/L)	5.38 ± 0.4 4.9 (3–17.7)	5.3 ± 0.95 5.2 (4.1–7.9)	0.07
Creatinine (umol/L)	82.77 ± 21.94 80.3 (49–134.3)	73.3 ± 9.6 75.7 (58.6–78)	0.3
Congenital heart diease	2 (5.6%)	0	-
Atrial septal defect, *n* (%)	7 (19.4%)	0	-
Ventricular septal defect, *n* (%)	1 (2.8%)	0	-
Persistent ductus arteriosus, *n* (%) Ebstein Anomaly with atrial septal defect, *n* (%)	4 (11.1%)	0	-
cc-TGA with ventricular septal defect, *n* (%)	1 (2.8%)	0	-
Pulmonary atresia, ventricular septal defect and MAPCA’s	3 (11.1%)	0	-
Tetralogy of Fallot	5 (13.9%)	0	-
Univentricular heart	13 (36.1%)	0	-

**Table 2 cells-13-01494-t002:** Biochemical and vascular parameters characteristics of the study population and the control group. Data are presented as the mean ± SD, median, and min-max value. AoSP—aortic systolic pressure; AoDP—aortic diastolic pressure; AoPP—aortic pulse pressure; AP—augmentation pressure; AI—augmentation index; FMD—flow-mediated dilatation; FVC—forced vital capacity; IMT—intima media thickness; PWV—pulse wave velocity.

	Cyanotic Patients (*n* = 36)	Control Group (*n* = 20)	Cyanotic Patients vs. Control Group (*p*-Value)
ADMA [ng/mL]	1.4 ± 0.5 1.5 (0.5–2.7)	0.4 ± 0.1 0.4 (0.2–0.7)	<0.001
E-selectin [ng/mL]	57.6 ± 15.4 56.3 (36–87)	15.5 ± 8.6 13 (8.6–39.7)	<0.001
AoSP [mmHg]	110.44 ± 15.787 107 (81–151)	104.5 ± 7.1 103 (94–120)	0.3
AoDP [mmHg]	73.17 ± 11.37 72 (52–107)	71.7 ± 7.5 72 (57–86)	0.4
AoPP [mmHg]	37.31 ± 11.11 35 (19–74)	32.9 ± 7.2 34 (22–48)	0.6
AP [mmHg]	10.14 ± 7.3 9.5 (1–31)	5.3 ± 3.3 4.5 (1–12)	0.002
Al [%]	24.75 ± 13.49 27 (2–60)	15.7 ± 8.8 15 (3.8–34)	0.01
PWV (m/s)	7.4 ± 2.07 7.1 (4.4–14.5)	7 ± 0.9 6.9 (5.4–8.5)	0.6
IMT (mm)	0.06 ± 0.02 0.06 (0.04–0.11)	0.05 ± 0.01 0.05 (0.04–0.08)	0.3
FMD (%)	9.04 ± 5.61 7.2 (0.5–25)	8.4 ± 4.1 7.9 (0.4–16.1)	0.6

**Table 3 cells-13-01494-t003:** Baseline characteristics of the cardiopulmonary test parameters of the study population and the control group. Data are presented as the mean ± SD, median, and min-max value. FEV1—forced expiratory volume in the first second; FVC—forced vital capacity; HR—heart rate; VE/VCO3—the minute ventilation/carbon dioxide production; VO2—respiratory oxygen uptake.

	Cyanotic Patients (*n* = 36)	Control Group (*n* = 20)	Cyanotic Patients vs. Control Group (*p*-Value)
FVC	3.4 ± 1.1 3.4 (1.3–5.5)	4.5 ± 0.7 4.6 (2.7–6.7)	<0.001
FVC %	86 ± 16.7 90 (41–116)	98.2 ± 8.8 98.2 (81–112.1)	0.04
FEV1	2.5 ± 0.8 2.6 (0.9–4.1)	3.6 ± 0.6 3.6 (2.3–5.3)	<0.001
FEV1%	76.4 ± 19.4 77 (42–116)	97.9 ± 9.1 97.8 (82.9–115)	<0.001
HR	85.5 ± 14.7 85 (59–115)	79.2 ± 12.9 78.5 (60–110)	0.6
VE/VCO3	43.0 ± 12.7 46 (31.2–62.1)	27.7 ± 4.5 26.4 (21.8–38.8)	<0.001
VO2 [mL/kg/min]	9.95 ± 8.2 3.9 (1.3–25.6)	34.4 ± 6.5 13.6 (7.2–22.6)	<0.001
VO2 [%]	48.2 ± 16.7 49 (22–84)	117.95 ± 18.9 116 (78–151)	<0.001

**Table 4 cells-13-01494-t004:** Correlation between patients’ clinical, vascular, and cardiopulmonary exercise test parameters and e-selectin and ADMA concentration. AoSP—aortic systolic pressure; AoDP—aortic diastolic pressure; AoPP—aortic pulse pressure; AP—augmentation pressure; AI—augmentation index; FEV1—forced expiratory volume in the first second; FMD—flow-mediated dilatation; FVC—forced vital capacity; HDL—high-density lipoprotein; HR—heart rate; IMT—intima media thickness; LDL—low-density lipoprotein; PWV—pulse wave velocity; RBC—red blood cell count; VE/VCO3—the minute ventilation/carbon dioxide production; VO2—respiratory oxygen uptake.

	E-Selectin	ADMA
Age (years)	r = 0.3 *p* = 0.8	r = 0.4 *p* = 0.02
Oxygen blood saturation (%)	r = −0.4 *p* < 0.01	r = −0.05 *p* = 0.8
Hematocrit (%)	r = 0.6 *p* < 00.1	r = 0.2 *p* = 0.2
Hemoglobin (mmol/L)	r = 0.6 *p* < 0.01	r = 0.3 *p* = 0.1
RBC (10^12^/L)	r = 0.6 *p* < 0.01	r = 0.2 *p* = 0.4
Vascular and endothelial parameters		
AoSP [mmHg]	r = 0.2 *p* = 0.2	r = −0.01 *p* = 0.9
AoDP [mmHg]	r = −0.005 *p* = 0.98	r = −0.2 *p* = 0.1
AoPP [mmHg]	r = 0.3 *p* = 0.1	r = 0.03 *p* = 0.9
AP [mmHg]	r = 0.3 *p* = 0.05	r = 0.01 *p* = 0.9
Al [%]	r = 0.3 *p* = 0.07	r = 0.05 *p* = 0.8
PWV (m/s)	r = −0.1 *p* = 0.5	r = −0.2 *p* = 0.2
IMT (mm)	r = −0.2 *p* = 0.4	r = −0.2 *p* = 0.3
FMD (%)	r = 0.1 *p* = 0.5	r = 0.2 *p* = 0.3
Cardiopulmonary excericse test parameters		
FVC	r = −0.4 *p* = 0.03	r = −0.2 *p* = 0.4
FEV1	r = −0.5 *p* = 0.01	r = −0.09 *p* = 0.7
HR	r = 0.03 *p* = 0.8	r = 0.3 *p* = 0.095
VE/VCO3	r = 0.5 *p* = 0.003	r = 0.1 *p* = 0.5
VO2 [mL/kg/min]	r = −0.4 *p* = 0.02	r = 0.02 *p* = 0.9
VO2 [%]	r = −0.4 *p* = 0.04	r = −0.1 *p* = 0.6

## Data Availability

Not applicable.

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
