# Peer review of "E-Selectin and Asymmetric Dimethylarginine Levels in Adult Cyanotic Congenital Heart Disease: Their Relation to Biochemical Parameters, Vascular Function, and Clinical Status"

_cells, 2024, doi:10.3390/cells13171494_

Round 1

Reviewer 1 Report

Comments and Suggestions for Authors

This is an overall very well written paper that furthers out understanding of  E-selectin and ADMA levels in adult cyanotic congenital heart 2 disease. I have only one concern  about the methods. 

I know the authors are trying to avoid plagiarism but they should include more methods for The flow-mediated dilatation (FMD) and intima-media thickness (IMT) measure-112 ments.

I believe the paper should be accepted for publication in the journal cells following minor revisions. 

Author Response

  1. I know the authors are trying to avoid plagiarism but they should include more methods forThe flow-mediated dilatation (FMD) and intima-media thickness (IMT) measure-112 

Authors response: The method has been added.  

Reviewer 2 Report

Comments and Suggestions for Authors

Thirty-six patients (17 males and 19 females) with cyanosis were selected because they are affected by congenital cyanogen heart disease (CHD).

Patients aged from 20 to72 (mean age 42.3 +/-16.3 years) with less than 92% arterial blood oxygenation were enrolled. All patients were under the care of the Adult Congenital Heart Disease Outpatient Clinic. The control group encompassed 20 healthy individuals (10 females) aged 19-64 (mean age 38.2 years + / - 6 years).  The baseline characteristics of all included individuals are displayed in tables 1, 2 and 3. The exclusion criteria were the coexistence of Down syndrome, acute and chronic inflammatory disease (in the preceding 3 months), concomitant malignancies, diabetes mellitus, and smoking cigarettes.

All thirty six patients underwent clinical examination, blood testing (including demetylarginine and e -selectin) and cardiopulmonary tests. Endothelial function was assessed using intima-media thickness and flow-mediated dilatation. Vascular function using applanation tonometry methods was completed while considering aortic systolic pressure, aortic pulse pressure, augmentation pressure, augmentation index, pulse pressure amplification, and pulse wave velocity.

The results suggest a relationship between CHD and an increased plasma concentrations of both demetylarginine and e -selectin. These two substances seemed to be positively correlated with CHD: in particular with red blood cell, hemoglobin concentration, hematocrit and augmentation pressure.  Both demetylarginine and e-selectin were negatively correlated with blood oxygen saturation, forced expiratory one-second volume, forced vital capacity, and oxygen uptake. Only demetylarginine levels were found to be correlated with age.

Increased concentration of demetylarginine is explained by more prolonged exposure of the endothelium to inflammatory factors damaging. Of note, did you check the eritrocyte sedimentation rate, C reactive protein, transaminases, intercellular adhesion molecule 1 (ICAM1) and vascular adhesion molecule (VCAM)?

It has to be clarified that the low extent of your sample does not allow to draw any definitive conclusions.

Thanks a lot

Author Response

  1. Of note, did you check the eritrocyte sedimentation rate, C reactive protein, transaminases, intercellular adhesion molecule 1 (ICAM1) and vascular adhesion molecule (VCAM)?

Authors response: We did analyze C reactive protein, transaminases and ICAM 1  between the control and study groups and we did not find any significant statistical differences

  1. It has to be clarified that the low extent of your sample does not allow to draw any definitive conclusions.

Authors response: we add such a statement in limitations.
